

# Incidence and risk of dialysis therapy within 30 days after contrast enhanced computed tomography in patients coded with chronic kidney disease: a nation-wide, population-based study

Yun-Ju Shih[1], Yu-Ting Kuo[1,2,3], Chung-Han Ho[4,5], Chia-Chun Wu[6,7] and Ching-Chung Ko[1,8]

[1] Department of Medical Imaging, Chi Mei Medical Center, Tainan, Taiwan
[2] Department of Medical Imaging, Kaohsiung Medical University Hospital, Kaohsiung, Taiwan
[3] Department of Radiology, Faculty of Medicine, College of Medicine, Kaohsiung Medical University, Kaohsiung, Taiwan
[4] Department of Medical Research, Chi Mei Medical Center, Tainan, Taiwan
[5] Department of Hospital and Health Care Administration, Chia Nan University of Pharmacy and Science, Tainan, Taiwan
[6] Division of Nephrology, Department of Internal Medicine, Chi Mei Medical Center, Tainan, Taiwan
[7] Department of Pharmacy, Chia Nan University of Pharmacy and Science, Tainan, Taiwan
[8] Center of General Education, Chia Nan University of Pharmacy and Science, Tainan, Taiwan

Corresponding author
Ching-Chung Ko,
crazyboy0729@gmail.com

## ABSTRACT

**Background:** Patients with chronic kidney disease (CKD) are considered at risk of contrast-induced acute kidney injury and possible subsequent need for dialysis therapy. Computed tomography (CT) is the most commonly performed examination requiring intravenous iodinated contrast media (ICM) injection. The actual risk of dialysis in CKD patients undergoing CT with ICM remains controversial. Furthermore, it is also uncertain whether these at-risk patients can be identified by means of administrative data. Our study is conducted in order to determine the incidence and risk of dialysis within 30 days after undergoing contrast enhanced CT in CKD coded patients.

**Methods:** This longitudinal, nation-wide, populated-based study is carried out by analyzing the Taiwan National Health Insurance Research Database retrospectively. Patients coded under the diagnosis of CKD who underwent CT are identified within randomly selected one million subjects of the database. From January 2012 to December 2013, 487 patients had undergone CT with ICM. A total of 924 patients who underwent CT without ICM are selected as the control group. Patients with advanced CKD or intensive care unit (ICU) admissions are assigned to the subgroups for analysis. The primary outcome is measured by dialysis events within 30 days after undergoing CT scans. The cumulative incidence is assessed by the Kaplan–Meier method and log-rank test. The risk of 30-day dialysis relative to the control group is analyzed by the Cox proportional hazards model after adjusting for age, sex, and baseline comorbidities.

**Results:** The numbers and percentages of dialysis events within 30 days after undergoing CT scans are 20 (4.1%) in the CT with ICM group and 66 (7.1%) in the
CT without ICM group ($p$ = 0.03). However, the adjusted hazard ratio (aHR) for 30-day dialysis was 0.84 (95% CI [0.46–1.54], $p$ = 0.57), which is statistically non-significant. In both advanced CKD and ICU admission subgroups, there are also no significant differences in 30-day dialysis risks with the aHR of 1.12 (95% CI [0.38–3.33], $p$ = 0.83) and 0.95 (95% CI [0.44–2.05], $p$ = 0.90), respectively.

**Conclusions:** Within 30 days of receiving contrast-enhanced CT scans, 4.1% of CKD coded patients required dialysis, which appear to be lower compared with subjects who received non-contrast CT scans. However, no statistically significant difference is observed after adjustments are made for other baseline conditions. Thereby, the application of administrative data to identify patients with CKD cannot be viewed as a risk factor for the necessity to undergo dialysis within 30 days of receiving contrast-enhanced CT scans.

## INTRODUCTION

Administration of intravenous iodinated contrast media (ICM) during computed tomography (CT) scans plays an indispensable role in improving diagnostic performance in numerous settings and is thereby widely employed in clinical practice. The healthcare industry has seen a steady increase in demand for CT scans ordered by clinicians, which inevitably leads to a corresponding increase in use of ICM (*Hu et al., 2016*). However, intravenous ICM also raises major concerns of the development of contrast-induced acute kidney injury (CI-AKI). The exact pathophysiology leading to this condition is poorly understood, but could possibly result from contrast-induced vasoconstriction, a decrease in filtration via tubuloglomerular feedback, direct renal epithelial toxicity, and increased free radial formation (*Pasternak & Williamson, 2012*). The elevations of blood osmolality and tubular fluid viscosity may also cause microvascular thrombosis and renal tubular obstruction, respectively (*Mehran, Dangas & Weisbord, 2019*). After excluding other etiologies of renal impairment, the diagnostic criteria for CI-AKI are serum creatinine elevation varying from 0.3 to 0.5 mg/dL in an absolute increase, or a 25–50% relative increase within the time period of 24–72 h after contrast administration (*McDonald et al., 2013*; *Mehran & Nikolsky, 2006*; *Meinel et al., 2014*).

It is known that patients with chronic kidney disease (CKD) are particularity at risk of developing CI-AKI due to diminished renal function reserve (*Davenport et al., 2013*; *Kim et al., 2010*; *ACR Committee on Drugs & Contrast Media, 2018*; *Mehran & Nikolsky, 2006*; *Scharnweber, Alhilali & Fakhran, 2017*). However, there is also emerging evidence suggesting that the risk of CI-AKI in CKD patients may have been overestimated in the past (*Garfinkle, Stewart & Basi, 2015*; *Hinson et al., 2017*; *McDonald et al., 2013*; *Mehran, Dangas & Weisbord, 2019*; *Van Der Molen et al., 2018*). The type, amount and route of ICM given to a patient are also crucial parameters. Patients receiving high-osmolar contrast medium (HOCM), i.e., ionic monomers, large volumes of ICM administration, or intraarterial ICM injections are at high risks of developing CI-AKI (*Pasternak &*

*Williamson, 2012*), and these findings are mostly based on studies focusing on coronary interventional procedures. However, the doses given during contrast-enhanced CT scans are intravenous and usually smaller in volumes as compared with coronary interventional procedures. In addition, the majority of ICM now used in modern clinical practices are low-osmolar (LOCM) and iso-osmolar contrast media (IOCM). Therefore, many are now questioning the existence of clinically significant CI-AKI in this setting (*Mehran, Dangas & Weisbord, 2019*; *Pasternak & Williamson, 2012*). Baseline fluctuations in renal function indices may also cause spurious indications of CI-AKI, which are unrelated to ICM administration (*Bruce et al., 2009*). Nevertheless, it is the current consensus amongst practicing radiologists that pre-existing CKD is still the most important risk factor in the development of CI-AKI (*ACR Committee on Drugs & Contrast Media, 2018*; *Van Der Molen et al., 2018*). In addition, the prevalence of CKD is high in many countries (*Hill et al., 2016*). If the risk is truly overestimated, contrast-enhanced CT scans may be inappropriately withheld from patients with legitimate clinical indications, resulting in suboptimal or delayed diagnoses of serious conditions and unfavorable clinical outcomes.

Contrast-induced acute kidney injury is often but not always a reversible process (*Scharnweber, Alhilali & Fakhran, 2017*). One of the adverse events after CI-AKI causing the greatest concern is the possible necessity for dialysis therapy due to sustained renal function impairment. The exact figure for CKD patients who require dialysis shortly after undergoing CT with ICM is not well known, and it possess a high level of clinical interest. The necessity for dialysis in the short-term is defined within a time-frame of 30 days after undergoing CT scan, in concordance with other studies published in the literature (*McDonald et al., 2014*, *2015*, *2017a*). To directly identify CKD patients, it is essential to obtain laboratory information. An alternative method is to select patients with CKD based on diagnostic codes within an administrative dataset. The usefulness for administrative data regarding CKD status has been validated in various research articles, which reveal inconsistent sensitivities but very high specificities (*Grams et al., 2011*; *Navaneethan et al., 2011*; *Ronksley et al., 2012*; *Tonelli et al., 2015*). In the published literature, numerous studies have been conducted implementing diagnostic codes to identify CKD patients (*Chang et al., 2014*; *Chen et al., 2016*; *Chung et al., 2017*; *Yu et al., 2017*). In this study, a longitudinal, nation-wide, population-based data is used to evaluate the incidence and risk of the necessity for dialysis within 30 days of undergoing contrast-enhancing CT scans amongst CKD patients. It is also our imperative to determine whether the administrative data is able to successfully identify CKD patients who are at risk.

## MATERIALS AND METHODS

### Study design and setting

The National Health Insurance (NHI) program in Taiwan, launched in March of 1995, is a compulsory single-payer social insurance plan aimed at delivering healthcare to all citizens. It covers nearly all of the 23 million people living in Taiwan. The National Health Insurance Registry Database (NHIRD) is a database containing comprehensive information such as demographic data, diagnostic codes, drug prescription, and procedure codes after encryption of patients' personal information. For research purposes, a subset of

database, namely, the Longitudinal Health Insurance Database 2010 (LHID2010), was provided by the National Health Research Institutes, and it had been implemented in numerous research articles (*Chang et al., 2015*; *Chou et al., 2015*; *Ho et al., 2014*; *Wu et al., 2014*). This dataset contains of one million individuals chosen randomly from the NHIRD at the year 2010 and contains medical information from January 1997 to December 2013 of each subject under the coverage. The Catastrophic Illness Registry Data file was supplemented for identification of deaths during the study period. The Institutional Review Board in the Chi Mei Medical Center passed approval of this study (No: 10706-E03) and the requirement of informed consent was waived.

## Patient identification and selection

Within the LHID2010, CKD patients were identified by using relevant International Classification of Diseases, Ninth Revision, Clinical Modification (ICD-9-CM) codes with at least two outpatient claims or one inpatient claim documented in the diagnosis codes section. Patients receiving CT scans were selected by using the Taiwan NHI codes for procedures, including information on study date and the presence or absence of ICM administration. The specific ICM injected was unknown, however, the majority of ICM used in Taiwan is LOCM, i.e., non-ionic monomers. CKD patients receiving CT without ICM were assigned to the control group. The specific dates of CT scans were not available before year 2012 within LHID2010, therefore only CKD patients receiving CT during January 2012 to December 2013 were included for analysis. The following patients were excluded: ages below 20 or over 100 years old; patients who were already receiving regular dialysis before the date of the CT scan; patients receiving CT scans for two or more times within 30 days; and patients who had undergone angiography or transarterial embolization within 30 days of the CT scan. The flow diagram for patient selection was shown in Fig. 1. All included patients were followed up for a 30-day period, to the occurrence of dialysis or death events, until the end of December 2013 or when leaving the NHI program. The follow-up period was not limited to 30 days when evaluating the time interval from CT to dialysis event as a secondary outcome described subsequently. Furthermore, since laboratory data was not available in LHID2010, the severity of CKD could not be directly assessed. An indirect approach was to select CKD patients receiving erythropoietin-stimulating agents (ESA) as the advanced CKD subgroup in this study. The NHI reimbursed ESA only in CKD patients with serum creatinine above 6 mg/dL, approximately equivalent to an estimated glomerular filtration rate (eGFR) <15 mL/min/1.73 $m^2$, and a hematocrit level below 28% observed concurrently. This approach to subdivide patients with advanced CKD had also been used in previous research articles (*Chung et al., 2017*; *Hsieh et al., 2016*). Another subgroup was the status of admission to the intensive care unit (ICU) during the time the CT scan was performed.

## Assessment of comorbidities and confounding factors

Data including age, sex, and comorbid conditions were collected and incorporated into the statistical analyses. The age was stratified into <60 years old, 60–80 years old, and >80 years old groups. Baseline hypertension (HTN), diabetes mellitus (DM), ischemic

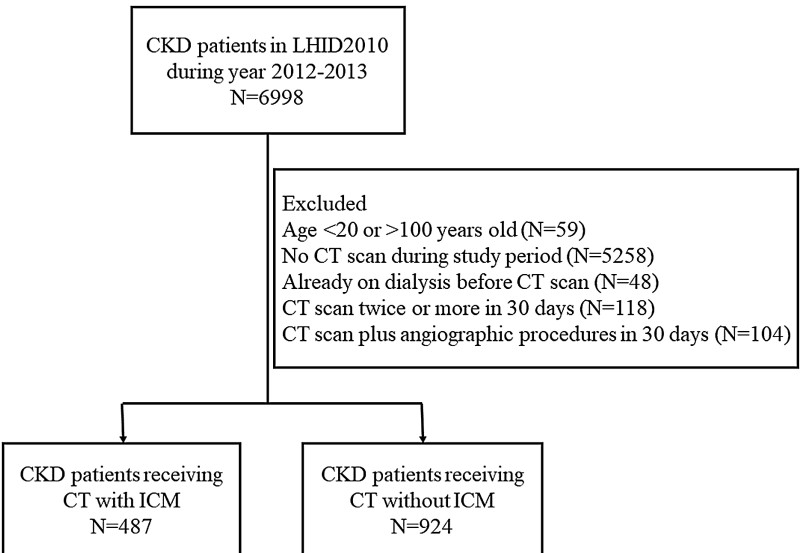

**Figure 1 Flow diagram for patient selection.** Initially, 6,998 CKD patients were identified in the studied dataset. Some patients were excluded due to the reasons listed, and there were 1,411 patients included for analysis. Subjects were divided into two groups, CT with ICM (487 patients) and CT without ICM (924 patients). LHID2010, Longitudinal Health Insurance Database 2010 (a dataset derived from National Health Insurance Research Database in Taiwan); ICM, iodinated contrast media.

heart disease (IHD), peripheral arterial occlusive disease (PAOD), congestive heart failure (CHF), liver cirrhosis, hyperlipidemia, and anemia were recorded based on the ICD-9-CM codes before the date of the CT scan was performed.

## Outcomes measurement

The primary outcome was measured by the occurrence of dialysis events within 30 days of CT scans, which was recognized by Taiwan NHI codes for procedures. The secondary outcomes were dialyses within 30 days of CT scans in the two selected subgroups, dialysis in CKD excluding ICU patients, death within 30 days, and the time interval of dialysis therapies after CT scans in the advanced CKD subgroup.

## ICD-9-CM codes and Taiwan NHI codes for procedures

Chronic kidney disease patients were identified by using the ICD-9-CM codes and include the followings: 585 (CKD), 582 (chronic glomerulonephritis), 403 (hypertensive CKD), and 404 (hypertensive heart and CKD). Baseline comorbidities were also recognized with ICD-9-CM codes as the followings: HTN (401–405), DM (250, 357.2, 362.01, 362.02, and 366.41), IHD (411–414), PAOD (440–444), CHF (428), liver cirrhosis (571.2, 571.5, and 571.6), hyperlipidemia (272–272.4), and anemia (280–285). CT with ICM studies were searched by Taiwan NHI codes for procedures, both 33071B (CT with contrast) and 33072B (CT without and with contrast); CT without ICM studies were identified by Taiwan NHI codes for procedures 33070B (CT without contrast). Dialysis events were selected by using Taiwan NHI codes for procedures 58001C/58029C/58027C (hemodialysis) and 58018C (continuous veno-venous hemofiltration dialysis).

## Statistical analysis

Patient characteristics between the CT with ICM and CT without ICM groups, including age, sex, and comorbid conditions, were compared by using the Pearson's chi-square test for categorical variables. Kaplan–Meier analysis was performed to assess the cumulative incidence of dialysis between these two groups, and the log-rank test was used to measure the difference of incidence curves. The Cox proportional hazard model adjusted for age, sex, and comorbid conditions was used to demonstrate the risk ratio of dialysis and death between the two groups. Each of the baseline characteristic was viewed as a distinct dichotomous variable. Adjusted hazard ratios (aHRs) and 95% confidence intervals (95% CI) were calculated. The result for the time interval leading up to dialysis in advanced CKD subgroup was not following the Gaussian distribution; therefore, the median value, the first quartile, and the third quartile were reported. All statistical analyses were performed using Statistical Analysis System (SAS) statistical software (version 9.4; SAS Institute Inc., Cary, NC, USA). The Kaplan–Meier curves were plotted using STATA (version 12; Stata Corp., College Station, TX, USA). A two-tailed $p$-value of <0.05 was considered statistically significant.

## RESULTS

### Baseline characteristics of the study participants

There were 1,411 eligible CKD patients who underwent CT studies, including 487 patients receiving CT with ICM and 924 patients receiving CT without ICM. Patient characteristics are listed in Table 1. In the CT with ICM group, patients were younger, more of them were male, and a greater number of subjects had liver cirrhosis. On the other hand, there were more patients with HTN, DM, hyperlipidemia, and anemia in the CT without ICM group. We identified 99 patients with advanced CKD who were undergoing ESA therapy: 15 of them received CT with ICM and 84 patients received CT without ICM. There were also 227 patients (16.1%) staying in the ICU, and a greater number of these subjects underwent CT without ICM.

### Necessity for dialysis and death within 30 days of CT

There were 20 patients (4.11%) in the CT with ICM group and 66 patients (7.14%) in the CT without ICM group requiring dialysis within 30 days of CT scans. The cumulative incidences of dialysis events were drawn in Fig. 2. A greater number of patients underwent dialysis therapy in the CT without ICM group ($p = 0.0295$). However, no statistically significant difference was found after the adjustment of age, sex, and the specified comorbid conditions (aHR 0.84, 95% CI [0.46–1.54], $p = 0.5700$). A total of 16 patients (3.29%) died in the CT with ICM group, which was also not significantly different from the 61 patients (6.60%) who died in the CT without ICM group (aHR 0.62, 95% CI [0.35–1.10], $p = 0.1012$). In the advanced CKD and ICU admission subgroups, neither dialysis within 30 days after CT nor death rates differ significantly between the CT with or without ICM groups. After excluding the critically-ill patients in the ICU, the percentages of patients requiring dialysis in 30 days was 1.61% in the CT with ICM

**Table 1 Baseline characteristics of the studied CKD patients.**

| | CT with ICM<br>N = 487 (%) | CT without ICM<br>N = 924 (%) | p-Value |
|---|---|---|---|
| Age group, years | | | |
| <60 | 146 (29.98) | 171 (18.51) | <0.0001* |
| 60–80 | 238 (48.87) | 466 (50.43) | |
| ≥80 | 103 (21.15) | 287 (31.06) | |
| Sex | | | |
| Female | 197 (40.45) | 432 (46.75) | 0.0236* |
| Male | 290 (59.55) | 492 (53.25) | |
| Comorbidities | | | |
| HTN | 401 (82.34) | 824 (89.18) | 0.0003* |
| DM | 267 (54.83) | 582 (62.99) | 0.0029* |
| IHD | 263 (54.00) | 528 (57.14) | 0.2587 |
| PAOD | 108 (22.18) | 221 (23.92) | 0.4621 |
| CHF | 113 (23.20) | 238 (25.76) | 0.2913 |
| Cirrhosis | 53 (10.88) | 47 (5.09) | <0.0001* |
| Hyperlipidemia | 273 (56.06) | 568 (61.47) | 0.0488* |
| ESA usage (advanced CKD) | | | |
| Yes | 15 (3.08) | 84 (9.09) | <0.0001* |
| ICU admission | | | |
| Yes | 51 (10.47) | 176 (19.05) | <0.0001* |

Notes:
ICM, iodinated contrast media; HTN, hypertension; DM, diabetes mellitus; IHD, ischemic heart disease; PAOD, peripheral arterial occlusive disease; CHF, congestive heart failure; ESA, erythropoietin-stimulating agents; ICU, intensive care unit.
* $p < 0.05$ and are considered statistically significant by Pearson's Chi-square test.

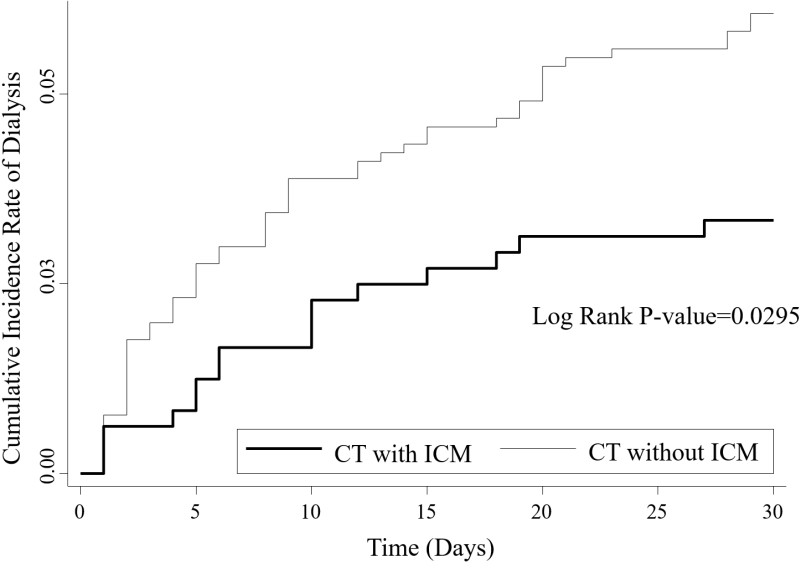

**Figure 2 Cumulative incidence of dialysis events within 30 days of CT scans in CKD patients.** The Kaplan–Meier curves revealed that the crude cumulative incidence of dialysis within 30 days of CT scans was higher in the CT without ICM group. ICM, iodinated contrast media.

**Table 2 Adjusted hazard ratios (aHR) for dialysis and death within 30 days after contrast enhanced CT scans in CKD patients, advanced CKD subgroup, ICU admission subgroup, and CKD patients excluding ICU admissions.**

|  | Outcome | CT with ICM N = 487 n/N (%) | CT without ICM N = 924 n/N (%) | Adjusted HR* (95% CI) | p-Value |
|---|---|---|---|---|---|
| CKD patients | 30-day dialysis | 20 (4.11) | 66 (7.14) | 0.84 [0.46–1.54] | 0.5700 |
|  | 30-day mortality | 16 (3.29) | 61 (6.60) | 0.62 [0.35–1.10] | 0.1012 |
| Advanced CKD subgroup^ | 30-day dialysis | 7 (46.67) | 30 (35.71) | 1.12 [0.38–3.33] | 0.8333 |
|  | 30-day mortality | 2 (13.33) | 4 (4.76) | 3.87 [0.16–92.07] | 0.4032 |
| ICU admission subgroup | 30-day dialysis | 13 (25.49) | 45 (25.57) | 0.95 [0.44–2.05] | 0.8973 |
|  | 30-day mortality | 6 (11.76) | 42 (23.86) | 0.84 [0.22–1.33] | 0.1788 |
| CKD patients excluding ICU admission | 30-day dialysis | 7 (1.61) | 21 (2.81) | 1.11 [0.40–3.06] | 0.8371 |
|  | 30-day mortality | 10 (2.29) | 19 (2.54) | 1.24 [0.54–2.85] | 0.6158 |

**Notes:**
n/N, event numbers/total numbers; ICM, iodinated contrast media; ICU, intensive care unit.
^ Advanced CKD subgroup is defined as CKD patients with concurrent usage of erythropoietin-stimulating agents.
* The HRs were adjusted by the age, sex, and comorbidities listed in Table 1.

**Table 3 Median time to dialysis in the advanced CKD subgroup after CT scans.**

|  | CT with ICM N = 15 | CT without ICM N = 84 |
|---|---|---|
| Dialysis events n/N (%) | 11 (73.33%) | 64 (76.19%) |
| Median time, days (Q1–Q3) | 12 (4–139) | 38 (4–184.5) |

**Note:**
n/N, event numbers/total numbers; Q1–Q3, interquartile range.

and 2.81% in the CT without ICM groups, respectively (aHR 1.11, 95% CI [0.40–3.06], p = 0.8371). The events, percentages and aHRs of dialysis and death are shown in Table 2.

## Time to dialysis in advanced CKD subgroup

The time interval leading up to dialysis in the advanced CKD subgroup is demonstrated in Table 3. A total of 75 patients (75.8%) required dialysis after CT. The median time interval leading to dialysis was 12 days (Q1–Q3: 4–139) in the CT with ICM group and 38 days (Q1–Q3: 4–184.5) in the CT without ICM group.

## DISCUSSION

In our study, the adjusted risks of dialysis and death within 30 days after CT scans in CKD patients were not increased, regardless of whether the patient received ICM or not. Similar results were reported in three retrospective studies from the same institution (*McDonald et al., 2014*, *2015*, *2017a*). Regardless of CKD stage 3, 4, or 5, the incidences of dialysis and death rates were also not affected by ICM (*Garfinkle, Stewart & Basi, 2015*; *McDonald et al., 2015*, *2017a*). Therefore, CT with ICM is probably not an independent risk factor for the necessity of dialysis and event of death in CKD patients, even in advanced stages, in the short-term observation. The crude dialysis rate was lower in our CT with ICM group, although not statistically significant after adjustment. One reason could

be the younger age and the existence of fewer comorbid conditions in this group. The crude dialysis rate elevation in CKD patients receiving CT without ICM was observed by others as well (*McDonald et al., 2017a*). A recent meta-analysis consisted of patients with acute ischemic stroke also revealed a lower CI-AKI risk in those receiving contrast enhanced CTs as compared with those who received non-enhanced ones (*Brinjikji et al., 2017*). Consistent with our results, they became non-significant after the adjustment of confounding factors. It is possible that various predisposing conditions lead to concerns of developing CI-AKI, and clinicians may have ordered non-contrast CTs instead. The underlying causes may have resulted in renal failure and subsequent elevation in the crude dialysis rate in our studied population.

Reported post-CT dialyses within 30 days were few in the reviewed studies, with an overall percentage of 0.2–0.3% in CKD patients and only 5.1% in CKD stages four to five patients (*McDonald et al., 2014*, *2017a*). However, our study revealed higher rates of post-CT dialyses within 30 days at 4.1–7.1%. When excluding the critically-ill ICU patients, the percentages were still high at 1.6–2.8%. This finding could be related to the fact that Taiwan has the highest incidence of end-stage renal disease (ESRD) in the world (*United States Renal Data System, 2017*). Abundant dialysis-related resources are available, and the NHI provides full coverage of incurred expenses. Due to this relatively common practice, doctors, patients, and their family members would probably prefer dialysis therapy in the occurrence of renal failure in most situations. Another possible cause is that some CKD patients were unlabeled from ICD-9-CM, especially those with less severe disease (*Ferris et al., 2009*). Therefore, the reported incidences of dialyses could be overestimated in our study.

The importance of critical illness was mentioned in a previous study, stating that ICU patients with eGFR <45 mL/min/1.73 m$^2$ demonstrated an increased risk of emergent dialysis within 7 days after contrast enhanced CT scans (*McDonald et al., 2017b*). An earlier article also demonstrated that the development of CI-AKI may result in increased need for dialysis in the ICU (*Valette et al., 2012*). However, in our results, the necessity for post-CT dialysis within 30 days and death rates did not differ in the ICU subgroup. A more detailed stratification of patient condition, especially any acute illness, might be required to reveal any undesirable effect of ICM injected during CT particularly in the critically-ill setting.

The advanced CKD subgroup appears interesting since they were presumed to be most susceptible to CI-AKI and the possibility of dialysis. It is noteworthy that one previous study mentioned that CKD patients who underwent exposure to ICM during CT scans more than once per year may develop ESRD earlier in the time scale of years (*Hsieh et al., 2016*). However, CKD patients with concurrent usage of ESA were excluded from their study due to proposed short-interval deterioration of residual renal function. Therefore, these pre-dialysis patients are the focus of our scrutiny in determining whether the time-interval to dialysis requirement after intravenous ICM exposure shortens. The result, however, included only small numbers of patients as the median time decreased from 38 to 12 days. It was inconclusive while containing wide interquartile ranges. Further

investigation is warranted to clarify whether contrast exposure truly shortened the time to dialysis requirement in advanced CKD patients.

This is the first national database study to describe CKD patients receiving CT with ICM is not associated with higher risk of dialysis in the short-term. The advantage of this study is the conduction of a nation-wide population-based design to minimize potential single institution related biases. Whether the risk of CI-AKI varies among different races remains unclear (*Chawla et al., 2017*; *Powell et al., 2018*), yet there are no previous articles aimed specifically at Asians before this current study. In addition, it is unknown if CKD patients identified from administrative data can be viewed as a risk factor in terms of dialysis requirement within 30 days before our present study.

Considering that the information was extracted from a national research database, the cohort size appeared smaller than expected; but it was also a result of our rigorous selection criteria. Even if, our results offered valuable information in clinical practice. Although the relatively small cohort size may decrease the statistical power and increase the risk of a false negative result, our case numbers of dialysis events to be analyzed were larger than previous comparable studies showing the same results (*McDonald et al., 2014*, *2015*).

This study still has several other limitations. The recognition of diagnoses including CKD and comorbid conditions were based on ICD-9-CM diagnostic codes, and cases of mis-registration cannot be excluded. The indications for CT scans were unknown, and acute medical illnesses were also not analyzed due to database restrictions. Details on the volume and specific ICM injected were not retrievable in our dataset, but the majority of ICM used were LOCM among routine clinical practice in Taiwan. HOCM has now become obsolete, and IOCM is rarely used. There is no sufficient evidence to suggest a significant CI-AKI risk difference between LOCM and IOCM (*Eng et al., 2016*), and our results will unlikely change when considering that few patients actually received IOCM. And lastly, this is a retrospective cohort study, which inevitably contains biases inherently related to the study design. For further investigation, a well-designed prospective cohort study in CKD patients comparing those receiving CT with or without ICM can be conducted. The type and volume of ICM, acute medical conditions, nephrotoxic medications, and preventive measurements should be controlled. Novel biomarkers to predict CI-AKI and dialysis therapy requirement could also be studied in the future.

## CONCLUSIONS

In our population-based study, the percentage of CKD patients who require dialysis within 30 days of undergoing contrast enhanced CT scans is 4.1%, a lower incidence compared with subjects who receive non-contrast enhanced scans. However, no statistically significant difference is observed after adjustments are made for other baseline conditions. Patients with CKD identified via the application of administrative data, therefore, do not appear to be at risk of dialysis within 30 days after CT with ICM studies.

### Funding

This work was supported by the Chi Mei Medical Center, Tainan, Taiwan (No. CMFHR10789). The funders had no role in study design, data collection and analysis, decision to publish, or preparation of the manuscript.

### Grant Disclosures

The following grant information was disclosed by the authors:
Chi Mei Medical Center, Tainan, Taiwan: CMFHR10789.

### Competing Interests

The authors declare that they have no competing interests.

### Author Contributions

- Yun-Ju Shih conceived and designed the experiments, performed the experiments, prepared figures and/or tables, authored or reviewed drafts of the paper, approved the final draft.
- Yu-Ting Kuo conceived and designed the experiments, authored or reviewed drafts of the paper.
- Chung-Han Ho performed the experiments, analyzed the data, contributed reagents/materials/analysis tools, prepared figures and/or tables.
- Chia-Chun Wu conceived and designed the experiments, contributed reagents/materials/analysis tools.
- Ching-Chung Ko conceived and designed the experiments, authored or reviewed drafts of the paper, approved the final draft.

### Ethics

The following information was supplied relating to ethical approvals (i.e., approving body and any reference numbers):

The Institutional Review Board of Chi Mei Medical Center approved this study (No: 10706-E03) and waived the requirement of informed consent.

### Data Availability

Raw data for this work was obtained by application from the NHIRD, Taiwan (http://nhird.nhri.org.tw/en/index.htm) and may not be shared according to the database's rules governing use. Access to the data used in this study may be obtained by citizens of the ROC who fulfill the requirements of conducting research projects.

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
