# Peer review of "Incidence and risk of dialysis therapy within 30 days after contrast enhanced computed tomography in patients coded with chronic kidney disease: a nation-wide, population-based study"

_PeerJ, doi:10.7717/peerj.7757_

## Round 0.1 · original submission · Major Revisions

The manuscript focuses on a topic of potential interest. However, as highlighted by both reviewers, the study has some major shortcomings that should be addressed. In particular, in addition to the others issues, the authors should consider i) the relatively small cohort size that raises concern about statistical inaccuracy because of deficient adjusting; ii) the need to discuss their findings in relation to those already available in the literature.

Reviewer 1 ·

Basic reporting

- Minor language improvements are necessary and rephrasing should be considered for example in lines 48, 66, 79-82 and 97.
- Your introduction should explain some more relevant aspects that are important for the reader. Please comment briefly on the pathophysiology of CI-AKI. Please comment on the role of contrast media type (osmolarity), volume and application modality (intraarterial (angiography) vs. intravenous) in your introduction. You might refer to: Mayo Clin Proc. 2012 Apr;87(4):390-402. doi: 10.1016/j.mayocp.2012.01.012.
- Thank you for providing the raw data. Please check that tables and figures are explained sufficiently in the accompanying text. E.g.:
o Table 2: You should add definitions of the subgroups. Especially “advanced CKD subgroup” is not understandable by itself.
o Figure 1 & 2: Please add a subtext to describe the figures and explain the abbreviations used.

Experimental design

- Considering the chosen methodology (data extraction out of a national research database), the cohort size is deficient.

Validity of the findings

- Please discuss your finding, that the incidence of CM-AKI is more frequent in CT without ICM (even though not statistically significant), in more detail. According to table 1, the population of CT with ICM was significantly younger and has significantly less comorbidities. Can you explain this? You should pick this up in your discussion.
- In your discussion you mention the lack of information concerning the used contrast media type and volume. Please discuss this limitation of your study in more detail and refer to studies comparing isoosmolar and hypoosmolar contrast media. See Ann Intern Med. 2016 Mar 15;164(6):417-24. doi: 10.7326/M15-1402. Epub 2016 Feb 2.
- Do you think the higher rate of AKI in the group of non-contrast CT could be because of “background fluctuation” of kidney function? See AJR Am J Roentgenol. 2009 Mar;192(3):711-8. doi: 10.2214/AJR.08.1413.
- Please give a final statement about the significance of your work in the context of this relevant topic that is discussed so contradictorily. Which problems should be addressed in future studies and how should these be performed?

Additional comments

Thank you for your interesting manuscript about this relevant and controversially discussed topic. In my opinion, the main difficulty of your work is the relatively small cohort size in consideration of your experimental design. This raises significant statistical inaccuracy because of deficient adjusting. This needs to be discussed straightforwardly, as mentioned above.

Reviewer 2 ·

Basic reporting

This manuscript by Shih Y et al examined the risk of dialysis therapy within 30 days of contrast enhanced CT in CKD patients. Overall, this study showed the dialysis and death risks within 30 days after CT scan in CKD patients are not increased irrespective of ICM.

Experimental design

Author collected data between 2012 and 2013. The CKD patients were divided into two groups. 1. 480 patients were CT with ICM, 2. 924 patients were CT without ICM. Outcome were measured using Kaplan-Meier survival analysis method and Cox proportional hazards model.

Validity of the findings

The rational and results are appreciated. It is also interesting to see in advanced CKD and ICU patients had no difference CT with or without.

Additional comments

1. How did the patients selected in each group? Randomly?
2. Table 1: Add seperate p-value for each age group.
3. There many study including large trials showing ICM leads to AKD. How could author justify beneficial and deleterious effect? How do they differ?
4. Is there difference in ethnicity or any other parameter matters?
5. Why within 30 days of CT? Is there any reason behind it?
6. Data set and analysis are convincing but based on previous study reported and even our own clinic patients who undergoes CT with ICM get AKI.
7. Method should be improved. How did data been collected
8. All the patients are over 60 years of age. Do they matter to adjust the age for cumulative incidence of dialysis events?

---

## Round 0.2 · Minor Revisions

The revised manuscript is definitely improved. However, a few minor issues remain to be addressed, as indicated by Reviewer 1. In particular, the authors should discuss more critically the major limitation of the study, namely the small cohort size.

Reviewer 1 ·

Basic reporting

As the article is of particular impact and up-to-date, you might consider to cite the recent review by Mehran et al in NEJM (N Engl J Med 2019; 380:2146-2155 )DOI: 10.1056/NEJMra1805256 )

Experimental design

You might consider discussing your major limitation (small cohort size), which can not be improved due to the chosen methods, more critically.

Validity of the findings

no comment

Additional comments

Thank you for your extensive revision work. Your manuscript has improved considerably now. In my eyes, your study has one major limitation, which is the small cohort size. As this limitation can not be resolved, critical discussion is essential.

Reviewer 2 ·

Basic reporting

'no comment'

Experimental design

'no comment'

Validity of the findings

'no comment'

Additional comments

The author addressed all the queries I have raised on the manuscript.

---

## Round 0.3 · accepted · Accept

The revised manuscript is definitely improved and we have no further comments.

Reviewer 1 ·

Basic reporting

no comment

Experimental design

no comment

Validity of the findings

no comment

Additional comments

Thank you for your revision. The manuscript has improved markedly and the limitations of the study are now sufficiently discussed.